# Effects of Fluctuating Thermal Regimes and Pesticides on Egg Hatching of a Natural Enemy *Harmonia axyridis* (Coleoptera Coccinellidae)

**Jingya Yu** [1,†], **Chong Li** [2,†], **Likun Dong** [1], **Runping Mao** [1], **Zhihua Wang** [1], **Zhangxin Pei** [1] **and Letian Xu** [2,*]

1    Institute of Plant Protection, Wuhan Institute of Landscape Architecture, Wuhan 430022, China
2    State Key Laboratory of Biocatalysis and Enzyme Engineering, School of Life Sciences, Hubei University, Wuhan 430062, China
*    Correspondence: letian0926@163.com
†    These authors contributed equally to this work.

**Abstract:** The harlequin ladybird, *Harmonia axyridis*, is a valuable asset in integrated pest management (IPM); however, issues related to low-temperature storage and transportation have resulted in low hatching rate, while the use of pesticides may lead to non-target effects against this natural enemy during field application. Fluctuating thermal regimes (FTR) have been shown to be beneficial during the low-temperature storage, and the type and concentration of insecticides used are crucial for field application of *H. axyridis*. Despite this, little research has been conducted on the effects of FTR on the hatching rate of ladybird eggs, and the impact of pesticides on their egg viability remains unclear. To address these gaps, we investigated the effects of different thermal temperatures, recovery frequencies (the number of changes in temperature conditions per unit time), and recovery durations (the duration of the treated temperature condition) on egg hatching under constant low-temperature conditions. We also examined the toxicity and safety of seven commonly used insecticides on egg hatching. Our results indicate that the temperature during FTR application did not significantly affect egg hatching, but the interaction between temperature and recovery frequency can significantly affect egg hatching. Moreover, the recovery frequency and recovery duration had a significant impact on hatching. Under specific conditions, the hatching rate of eggs subjected to FTR was similar to that of eggs stored at 25 °C. Furthermore, we found that matrine (a kind of alkaloid pesticide isolated from *Sophora flavescens*) had low toxicity to ladybird eggs and is a safe pesticide for use in conjunction with this natural enemy. The study provides valuable information on effectively managing *H. axyridis* by taking into account both storage temperature and pesticide exposure.

**Keywords:** fluctuating thermal regime; pesticide; *Harmonia axyridis*; egg hatching

## 1. Introduction

Insect pests have caused significant losses to agricultural production worldwide, and the situation is being exacerbated by the environment and climate change [1]. An effective insect pest management strategy is therefore crucial to minimize losses and maximize agricultural yields. Chemical pesticides have been widely used in pest management, but they have numerous negative impacts on the environment and human health and also lead to the development of pesticide resistance in target pests [2]. As a result, there is an urgent need for environmentally friendly alternatives to chemical pesticides and biocontrol is one of the most promising solutions [3]. Previous studies have demonstrated that the use of eco-friendly methods, such as employing natural enemies or entomopathogens in crop fields, could have complementary effects in reducing the population density of targeted insect pests [4,5]. Thus, by using natural enemies as a complementary resource, insect pest populations can be effectively decreased, reducing the use of pesticides and improving the number of biological interactions in a specific agriculture and agroforestry ecosystem [6].

The harlequin ladybird, *Harmonia axyridis* (Pallas) (Coleoptera Coccinellidae), is a species of ladybird originating from Asia that primarily feeds on aphids [7]. It is considered an effective biocontrol agent for aphids, which are destructive crop pests globally [8,9]. For example, an effective integrated pest management program was reported for managing pecan aphid, in which *H. axyridis* adults and other predators were released [10]. In China, *H. axyridis* is generally applied in the field by releasing egg cards [11]. Nevertheless, the eggs of *H. axyridis* are usually stored at low-temperature (normally 5 °C) before field application [12], which significantly decreases their survival rate and limits the application range and commercial mass production [13]. Fortunately, previous studies have shown that a short high temperature pulse (fluctuating thermal regime [FTR]) treatment during low-temperature storage of pupae could significantly increase adult emergence rate and survival [13,14]. Nevertheless, the effect varies with the frequency of the FTR [15,16]. However, it remains unknown whether the FTR treatment could increase the hatching rates of *H. axyridis* eggs and which frequency of FTR would be the most optimal.

In addition to the challenges in storage, the eggs of *H. axyridis* released in the field are also likely to come into contact with pesticides and their residues as they are often used in conjunction with other biocontrol methods and chemicals [17,18]. Recently, the negative effects of chemical pesticides on non-target insect have been well documented, including increased mortality, slowed growth and development, and reduced reproduction [19–21]. Not surprisingly, insecticides have a detrimental effect on the egg hatching of some non-target insects with varying levels of toxicity and have been shown to have a negative effect on the hatching rate of eggs in the ladybird *Eriopis connexa* (Germar) [22]. Although the impact of certain insecticides on the eggs of *H. axyridis* has been documented [7,17], the effect of other pesticides sprayed on eggs remains an open question.

In this study, we focused on the two major issues hindering the application of *H. axyridis* as a biocontrol agent; our objective was to assess the impact of different FTR and the toxicity of some common insecticides on the hatchability of *H. axyridis* eggs. We hypothesized that an optimal FTR treatment would result in an increased hatching rate, while the pesticides would have adverse effects on the eggs with varying levels of toxicity.

## 2. Materials and Methods

### 2.1. Insect Rearing and Egg Collection

Adult *H. axyridis* were collected from garden plants in Wuhan City for laboratory study and overwintering. The insects were reared on the wheat aphid, *Rhopalosiphum padi*, under controlled conditions of 25 ± 1 °C, 50–60% relative humidity, and a 16:8 (light:dark) photoperiod. For the experiment, newly laid eggs (<24 h) were collected on egg cards (6 mm × 3 mm) and used after three days.

### 2.2. Effects of Fluctuating Thermal Regimes on Egg Hatching Rate

The study employed 36 storage protocols and two groups. Specifically, a constant low-temperature (CLT) of 5 °C [23] and a constant high temperature (25 °C) were set as control groups. The treatment groups were defined by three fluctuating thermal regimes (FTRs) with distinct temperature profiles. Each FTR (15, 20, or 25 °C) was characterized by a CLT phase and a recovery phase with duration of 0.5 h, 1 h, 2 h, or 3 h, at specific recovery frequency (24 h, 48 h, and 72 h). All experiments were conducted under complete darkness [24,25]. Therefore, a total of 36 different protocol combinations were tested, resulting from the 3 (FTRs) × 4 (recovery durations) × 3 (recovery frequencies). The experiment included five replicates for each treatment, and in each replicate, we used a piece of egg card (~26 eggs on one card). After five days of treatment, egg cards were removed from the treatments, and transferred to a standard hatching protocol of 25 °C, constant darkness, and 70% relative humidity [26]. The number of incubated eggs was recorded daily for six days (Figure 1).

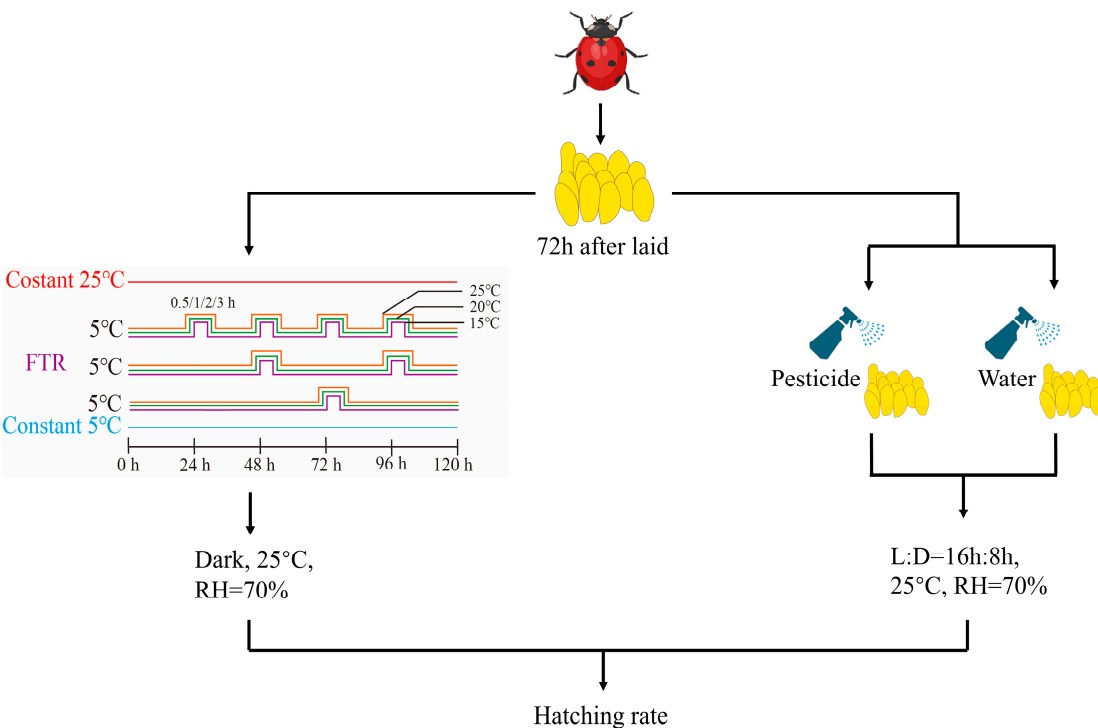

**Figure 1.** Workflow of FTR and pesticides effects on egg hatching.

### 2.3. Effects of Pesticides on Egg Hatching Rate

Seven pesticides were evaluated (Table 1). The pesticides and concentration were selected based on their current and potential use for controlling aphids and scale insects. Seven concentration levels were set for each tested pesticide, with distilled water used as solvent. The concentration levels were set in a doubling fashion, with each level being twice the concentration of the previous level. *H. axyridis* egg cards were exposed to the pesticides by topical application using spray bottles, simulating the conditions that eggs are likely to encounter in the field after pesticide application. The spray volume per application from the spray bottle was approximately 0.15 mL, and the distance of application was 5 cm. As a result, the liquid spread covered an area of approximately 17.16 cm$^2$ after spraying. Each treatment was replicated five times, with 20–30 eggs per replicate. After spraying, the eggs were kept at the same hatching condition as described in Section 2.2. The hatching assessments were conducted until no more larvae emerged (Figure 1).

**Table 1.** Insecticides tested for toxicity to the *Harmonia axyridis*.

| Insecticides | Formulation | Recommended Concentrations (mg/L) | Toxicity | Families | Initial Concentration (mg/L) | Final Concentration (mg/L) |
|---|---|---|---|---|---|---|
| 3% Emamectin benzoate | ME | 2000 | Low | Microbial pesticide | 125 | 8000 |
| 5% Abamectin | EW | 1000 | Moderate | Microbial pesticide | 50 | 3200 |
| 30% Thiamethoxam | SC | 666.67 | Low | Second generation nicotine | 62.5 | 4000 |
| 20% Acetamiprid | SL | 1000 | Low | Nicotine chloride compounds | 62.5 | 4000 |
| 20% Imidacloprid | SL | 1000 | Low | Neonicotinic insecticides | 84 | 5336 |
| 1.3% Matrine | EW | 1000 | Low | Alkaloid | 125 | 8000 |
| 0.3% Azadirachtin | SC | 1000 | Low | Biological insecticide | 125 | 8000 |

The pesticides were compared by the index of relative toxicity, which was calculated by dividing the $LC_{50}$ of a pesticide by the $LC_{50}$ of the 1.3% matrine EW [27]. Since the 1.3% matrine EW has the largest numerical value of $LC_{50}$, using it as a test agent for this calculation allows for toxicity comparisons between different pesticides. The safety of a pesticide to a natural enemy is indicated as relative safety factor [28]. The relative safety factor (Sr) was calculated by dividing the $LC_{95}$ of a pesticide by the recommended concentration of that pesticide for field application. The level of risk was divided into four grades, low risk means relative safety factor > 5, medium risk means 5 ≥ relative safety factor > 0.5, high risk means 0.5 ≥ relative safety factor > 0.05 [29].

*2.4. Statistical Analysis*

A three-way ANOVA analysis was conducted to assess the effects of recovery temperature, recovery duration, and recovery frequency, as well as their interactions on the egg hatching rate in order to identify the optimal storage conditions [30]. After that, the differences in hatching rate between every group were analyzed using analysis of variance (ANOVA) followed by Tukey's HSD tests. Toxicity analysis was conducted by using the log10 of pesticide concentration in each treatment as the independent variable (X) and the egg hatching failure rate as the dependent variable (Y). The probit regression model was used to fit the toxicity regression curve, and the $LC_{50}$ (median lethal concentration) and $LC_{95}$ (95% lethal concentration) values of the eggs were estimated by the probit mode. All statistical analyses were performed using IBM SPSS Statistics 20 software.

## 3. Results

### 3.1. The Influence of Fluctuating Thermal Regimes on the Hatching Rate of Eggs

There were no observed effects of temperature on hatching rate ($p = 0.461$, Table 2), suggesting that the temperature of the FTR has no statistical impact on egg hatching. Compared to temperature, the recovery frequency and recovery duration of the thermal fluctuations were significantly affected egg hatching (recovery frequency: $p = 0.027$; recovery duration: $p = 0.048$; Figure 2A, Table 2). Additionally, there was a significant interaction between recovery frequency and temperature on egg hatching, which warrants further analysis ($p = 0.007$; Figure 2B, Table 2).

**Table 2.** The correlation and interaction significance between the factors of FTRs on egg hatching rate.

| Factors | df | MSE | F | *p*-Value |
|---|---|---|---|---|
| Recovery frequency (1) | 2 | 0.229 | 3.722 | 0.027 |
| Temperature (2) | 2 | 0.048 | 0.778 | 0.461 |
| Recovery duration (3) | 3 | 0.166 | 2.698 | 0.048 |
| Between (1) + (2) | 4 | 0.225 | 3.661 | 0.007 |
| Between (1) + (3) | 6 | 0.067 | 1.091 | 0.370 |
| Between (2) + (3) | 6 | 0.105 | 1.713 | 0.122 |
| Between (1) + (2) + (3) | 12 | 0.057 | 0.929 | 0.520 |
| Error value | 144 | 0.062 | | |
| Total | 180 | | | |

The data were analyzed using the three-way analysis of variance (ANOVA) and Tukey's post hoc tests to detect the effects of the independent variables (recovery frequency, temperature and recovery duration) and their interactions on the egg hatching ($p = 0.05$). All statistical analyses were conducted using software. Statistically significant difference at $p < 0.05$.

The hatching rate of eggs under different FTRs are shown in Figure 2. The observed means, standard deviations, statistical differences, and ranges of temperature, recovery frequency, and recovery duration are given in Table 3. The hatching rate of *H. axyridis* eggs was high (99.00%) when incubated at a constant temperature of 25 °C, but significantly decreased to 54.29% at a constant temperature of 5 °C (Table 3, $p < 0.05$). The eggs stored under a 24-hour recovery frequency with a 20 °C temperature and a recovery duration of 1, 2, or 3 h showed a significantly higher hatching rate for *H. axyridis* than other treatments

(Figure 2, Table 3). Meanwhile, a negative effect on egg hatching rate was observed under long recovery duration (72 h) for each temperature (15, 20, or 25 °C) (Table 3). Although there was no statistically significant difference between some treatments, the egg hatching rate of *H. axyridis* tended to increase with longer recovery duration (Table 3).

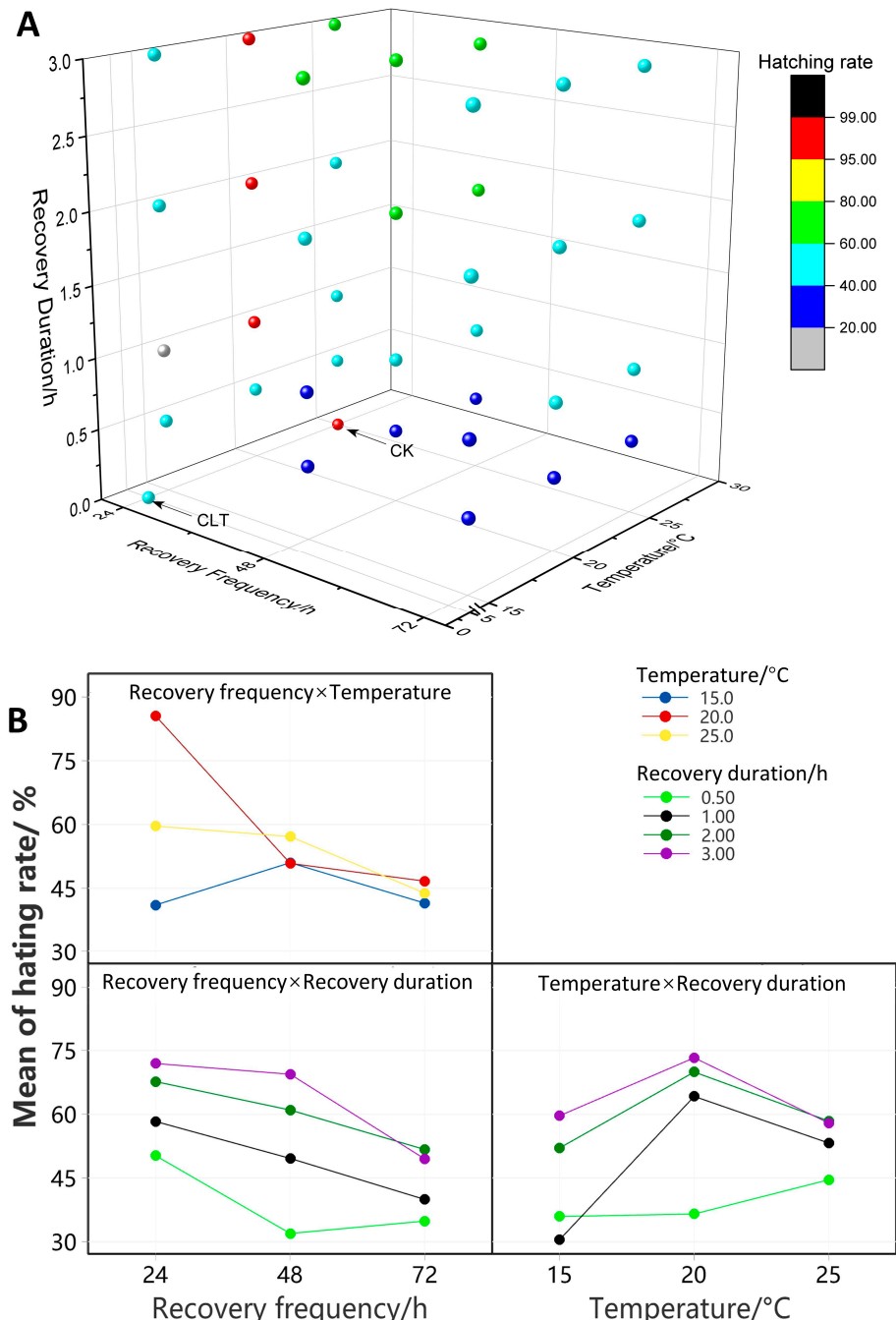

**Figure 2.** Egg hatching under different condition and interaction effects between factors. (**A**) Hatching rates of eggs exposed to different condition. Control eggs were maintained at a constant temperature of 25 °C or 5 °C. (**B**) Interaction effects plot showing the interactions between recovery frequency and temperature, recovery frequency and recovery duration, and temperature and recovery duration. Parallel lines indicate no interaction, while intersecting lines indicate the presence of an interaction. The degree of intersection reflects the strength of the interaction.

**Table 3.** The difference between the rate of egg hatching for *Harmonia axyridis* under different recovery frequency and recovery duration.

| Recovery Temperature | Recovery Frequency | Recovery Duration | | | |
|---|---|---|---|---|---|
| | | **0.5 h** | **1 h** | **2 h** | **3 h** |
| 15 °C | 24 h | 44.78 ± 7.34 | 19.69 ± 4.47 | 46.96 ± 13.80 a | 52.78 ± 20.10 |
| | 48 h | 33.55 ± 4.09 Ab | 38.33 ± 10.46 AB | 56.43 ± 1.62 Bb | 75.70 ± 3.48 AB |
| | 72 h | 29.45 ± 9.53 | 33.27 ± 8.51 | 52.77 ± 7.91 a | 50.45 ± 10.54 |
| 20 °C | 24 h | 48.86 ± 4.12 Aa | 97.49 ± 1.38 A | 97.30 ± 0.29 A | 98.41 ± 1.59 B |
| | 48 h | 25.18 ± 5.64 Ab | 52.81 ± 3.33 B | 61.49 ± 9.32 AB | 63.86 ± 1.76 B |
| | 72 h | 35.51 ± 8.42 a | 42.36 ± 4.11 | 51.09 ± 9.02 | 57.70 ± 5.79 |
| 25 °C | 24 h | 57.12 ± 3.90 A | 57.76 ± 14.60 AB | 58.85 ± 0.64 B | 64.70 ± 5.55 AB |
| | 48 h | 37.07 ± 2.14 | 57.69 ± 12.31 | 65.09 ± 7.67 | 68.75 ± 4.93 |
| | 72 h | 39.59 ± 3.67 | 44.20 ± 13.15 | 51.26 ± 0.67 | 40.28 ± 15.24 |
| Constant low temperatures | | 54.29 ± 7.52 a | | | |
| Control group | | 99.20 ± 0.72 b | | | |

Data (mean ± SE) followed by different letters indicate significant at the 0.05 level. Significant differences between different recovery duration under the same temperature and frequency are represented by capital letters; significant differences between different recovery frequencies under the same temperature and recovery duration conditions are represented by lowercase letters.

### 3.2. Effects of Seven Pesticides on the Hatching Rate of H. axyridis

A linear relationship between pesticide concentration and hatching rate was obtained, and the resulting linear regression equation and coefficient of determination were displayed in Table 4. Generally, the model fit the hatching rate distributions well, with an $R^2$ value of over 0.9. Based on the $LC_{50}$ of the pesticides on the eggs, 1.3% matrine EW was the least toxic compound ($LC_{50}$ = 10,752.25), while 20% acetamiprid SL showed the highest toxicity to eggs ($LC_{50}$ = 158.35) (Table 4). The toxicity of pesticides showed a similar pattern when based on $LC_{95}$ of pesticides on the eggs (Table 4). A comparison between groups sprayed with different pesticides showed a significant difference in toxicity based on the index of relative toxicity (index of relative toxicity > 1.5; Table 4) [27]. Among the pesticides, matrine showed the highest value of Sr (Sr = 10.75), while acetamiprid displayed the lowest Sr, indicating a high risk for *H. axyridis* (Sr = 0.16). Imidacloprid, abamectin, azadirachtin, and abamectin-aminomethyl exhibited a medium risk (Sr = 0.36, 0.56, 3.68), while thiamethoxam and matrine displayed a low risk (Sr = 10.45) for relative safety. (Table 4).

According to the relative safety factor, the risk level of pesticides to natural enemies is divided into four levels. Safety factor > 5 is considered low risk, 5 ≥ safety factor > 0.5 is considered moderate risk, 0.5 ≥ safety factor > 0.05 is considered high risk, and safety factor ≤ 0.05 is considered extremely high risk.

**Table 4.** Acute toxicity of the seven insecticides to eggs for *Harmonia axyridis*.

| Insecticide | Regression Equation | $LC_{50}$/mg/L | $LC_{95}$/mg/L | Correlation Coefficient | Relative Safety Factor | Index of Relative Toxicity |
|---|---|---|---|---|---|---|
| 20% acetamiprid SL | $Y = -1.411 + 0.009X$ | 158.35 | 342.89 | 0.914 | 0.16 | 67.90 |
| 20% imidacloprid SL | $Y = -4.362 + 1.708X$ | 358.15 | 3290.17 | 0.923 | 0.36 | 30.02 |
| 5% abamectin EW | $Y = -10.334 + 3.765X$ | 555.87 | 1520.19 | 0.948 | 0.56 | 19.34 |
| 0.3% azadirachtin SC | $Y = -5.406 + 1.516X$ | 3679.67 | 44,744.10 | 0.971 | 3.68 | 2.92 |
| 3% abamectin-aminomethyl ME | $Y = -8.773 + 2.339X$ | 5630.88 | 28,430.59 | 0.902 | 2.82 | 1.91 |
| 30% thiamethoxam SC | $Y = -4.687 + 1.219X$ | 6969.76 | 155,604.83 | 0.904 | 10.45 | 1.54 |
| 1.3% matrine EW | $Y = -4.870 + 1.180X$ | 107,52.25 | 263,162.094 | 0.911 | 10.75 | 1.00 |

Data for toxic regression equation, $LC_{50}$, and $LC_{95}$ were analyzed using probit analysis program.

## 4. Discussion

Insects, as poikilothermic animals, are greatly influenced by temperature, which plays a critical role in various biological aspects of insects [31]. These aspects include feeding behavior, survival, and morphological characteristics [32–34]. The effective rearing and preservation of beneficial insects, such as pest predators, pose challenges that need to be

addressed in practical production practices [35]. Eggs of *H. axyridis* are often stored at low temperatures during storage and transportation [36], which can negatively impact the hatching rate and larval development [37]. Similarly, we have previously observed a significant decrease in the hatching rate of *H. axyridis* eggs under low temperature storage conditions. FTR has been extensively employed to enhance the quality of eggs or pupae during the storage and transportation of natural enemies. However, it is important to note that the effects of FTR treatments may vary across different species and FTR factors, such as temperature and recovery frequency. We found that FTR treatment had a positive effect on the hatching of *H. axyridis* eggs exposed to low temperatures. Additionally, the FTR-based protocols with a temperature of 20 °C, recovery frequency of 24 h, and recovery duration of 1 h, 2 h, or 3 h showed significantly higher hatching rates compared to the control group.

According to the physiological recovery hypothesis, which is widely accepted to explain the effect of FTR on survival, insects benefit from periodic physiologically recovery to counteract the chilling injuries that accumulate during constant low temperature [38]. These recovery processes encompass various aspects, such as restoration of ion gradients, metabolic homeostasis, detoxification, and more [39]. Intriguingly, our study on *H. axyridis* revealed a negative effect associated with several types of FTRs, resulting in a lower egg hatching rate compared to the CLT group. This phenomenon was particularly prominent under high temperatures and long recovery frequency (refer to Figure 2 and Table 3). The reduced hatching rate of eggs may be attributed to unfavorable conditions that hinder or reverse the normal physiological repair process. Future research should aim to explore the specific pathways that lead to decreased egg hatching under high temperature and long frequency conditions. On the other hand, we revealed a novel role of different three factors and their interaction in FTR on egg hatching of *H. axyridis*. Specifically, we showed that the recovery frequency and recovery duration had a significant impact on egg hatching. While temperature alone did not show a statistically significant effect on egg hatching, we observed a significant interaction between recovery frequency and temperature in influencing egg hatching. This suggests that temperature alone may not directly affect egg hatching, but it does interact with frequency within a certain range to influence the process; however, exposure to high temperatures and long fluctuation frequency resulted in significantly lower hatching rates compared to the control. In the short term, FTR-based protocols could be a viable option for maintaining the stockpiling of biocontrol agents. However, given the limited duration of thermal time, the frequency of heating intervals, and the small scale of storage, future research should focus on exploring optimal conditions for storing *H. axyridis* eggs that are less dependent on specialized equipment, making them more practical for commercial growers [40].

It is vital to investigate the toxicity of pesticides on natural enemies so as to determine the specificity of pesticides for target or non-target insects and identify compounds that are less harmful to non-target insects [41]. Furthermore, this information is crucial for the implementation of integrated pest management, where the goal is to take advantage of the control effect of natural enemies while minimizing the impact on the non-target organisms [42]. Here, we found that the exposure to pesticides through surface contact resulted in decreased hatching rates and concentration-dependent mortality in *H. axyridis* eggs, similar to other Coleoptera species, such as *Harpalus pennsylvanicus* [43], *Coccinella undecimpunctata* [44], and *Serangium japonicum* [45]. In the study, we examined the impact of seven common pesticides on *H. axyridis* egg hatchability and evaluated their safety towards *H. axyridis*. All the tested pesticides had adverse effects on the *H. axyridis* eggs, with the hatched number varying according to pesticide concentration and type. Additionally, 20% acetamiprid SL with the lowest $LC_{50}$ was the most toxic compound, and this pesticide is relatively high-risk in terms of relative safety. matrine was found to be low risk for relative safety, which is suitable for applying with *H. axyridis*. The 30% thiamethoxam SC displayed a similar risk compared to the 1.3% matrine, making it a potential for integration with natural enemy. However, thiamethoxam exhibited higher toxicity compared to 1.3% matrine. Therefore, we maintain that the latter (1.3% matrine) is more appropriate for

application. Meanwhile, we found that the larvae hatching in insecticide-treated eggs had lower survival rates than larvae hatching in untreated eggs (unpublished data), possibly due to residual pesticides that remained in the larvae and lead to negative impacts [46–48]. Hence, a future direction of study should be to detect the impact of pesticide residues in larvae hatching from pesticide-treated eggs and to assess the combination of natural enemies and pesticides in another aspect.

Taken together, in this study, we aimed to examine the effect of fluctuating thermal regimes (FTRs) and seven common pesticides on the hatching rate of *H. axyridis*. The research presented here reinforces that FTR is a versatile tool with many potential benefits including long-term storage with high rate of hatching of insects in scientific collections. One possible application of FTR-based protocols is the small-scale storage of biocontrol agents for production [49]; however, to effectively store larger volumes of insects, FTR protocols require further refinement. By exploring storage conditions and suitable pesticide types and concentrations, our findings provide a theoretical guidance for the integration of pesticides with the biological control method. For example, when we have identified the varied toxicity of different pesticides towards predator eggs, we can prioritize the use of pesticides with lower toxicity to predator eggs in integrated pest management strategies, where predators and pesticides are used together for pest control. Nonetheless, research about determining the potential impact of FTRs on pesticide resistance is needed. Consequently, this study provides valuable information on the selective toxicity of pesticides to non-target insects, and highlights the importance of considering both temperature conditions and pesticide exposure in the management of *H. axyridis*.

**Author Contributions:** Conceptualization, J.Y. and L.X.; Methodology, J.Y. and L.X.; Software, C.L., R.M. and Z.W.; Validation, C.L. and L.X.; Formal analysis, J.Y.; Investigation, J.Y., C.L., L.D., R.M., Z.W. and Z.P.; Resources, J.Y.; Data curation, L.D. and Z.P.; Writing—original draft, C.L.; Writing—review & editing, C.L. and L.X. All authors have read and agreed to the published version of the manuscript.

**Funding:** This research was supported by National Natural Science Foundation of China (31971663), the Natural Science Foundation of Hubei Province (2022CFA061) and the Young Elite Scientists Sponsorship Program by CAST (2020QNRC001).

**Data Availability Statement:** All data are included in the manuscript.

**Conflicts of Interest:** The authors declare no conflict of interest.

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
