# Peer review of "Effects of Fluctuating Thermal Regimes and Pesticides on Egg Hatching of a Natural Enemy Harmonia axyridis (Coleoptera Coccinellidae)"

_agronomy, doi:10.3390/agronomy13061470_

Round 1

Reviewer 1 Report

Lines 11 and 16: "Harlequin ladybird" should not be italicized. Also, use lowercase instead of uppercase "h". Include the scientific name (in italics) following the common name (not in italics);

Line 18: "thermal temperatures" is pleonasm. Replace that with "temperatures". Also, specify that the "frequencies" and "durations" that you described are related to the temperature fluctuation;

Lines 20-22: This statement is wrong. There was an interaction between temperature and fluctuation frequency. Hence, temperature did play an important role;

Line 23: Specify the formulation of matrine that was tested;

Line 26: Replace "temperature conditions" with "storage temperature";

Line 27: Replace "axyridi" with "axyridis";

Line 51: Replace "cards(Yu and Liu, 2019)" with "cards (Yu and Liu, 2019).";

Line 56: Replace "storage could" with "storage of pupae could";

Line 57: Replace "increases the emergence rate of pupa and survival rate of adults" with "increase adult emergence rate and survival";

Line 52: Replace "that applied" with "released";

Line 68: Delete the first "the"; replace "hatching rate of eggs" with "egg hatching";

Line 69: Replace "non-targeted" with "non-target";

Line 69: Please, specify which insecticide you are referring to;

Line 72: Insert a space between "reported" and the citation;

Line 73: Replace "these" with "the"; replace the first "the" with "these";

Lines 72-72: The text misleads the reader to expect experiments on spraying methods, when in reality the study delivers the impact of insecticides applied with a single method, just like other previous studies;

Lines 73-74: Replace "the hatching rate of H. axyridis" with "the egg hatching". 

Line 85: Replace "eggs laid on the same day" with "newly laid eggs (<24 h)";

Line 86: Replace "on the third day after they were laid" with "three days later";

Line 88: From my point of view the constant low temperature (5 ºC) as well as the constant high temperature (25 ºC) are not protocols but they are controls instead. You are actually testing 36 different protocols composed of 3 fluctuating temperatures (15, 20, and 25 ºC), 4 durations (0.5h, 1h, 2h, and 3h), and 3 frequencies (24h, 48h, and 72h), i.e., 3x4x3=36;

Line 89: Delete the extra ")";

Lines 90-91: Please, explain what are these recovery durations (are they exposure times?), how they were applied, and why the "complete darkness";

Line 92: Explain what is the "thermophase recovery frequency";

Line 95: How many eggs in each card? You say "every five days", but for how many days total?

Lines 95-96: How many cards were removed from the treatments? And how did that happen since there was only one egg card per replicate? Please, provide more details.

Lines 88-98: This whole section is confusing and needs greater improvement.

Fig. 1. This figure is great but its left part (the one that explains the FTRs) leads to think that the experiment was done for 120h only, which is probably incorrect since the text describes egg card removals each 5 days. Hence, this experiment was carried out for a stretch of time that is not described anywhere in the methods but seems to be much more than 120 h. I suggest adjusting the figure to better describe the entire timeframe of the experiment. Also, the font size for the FTR part of this figure is ridiculously small. Please make it bigger. At the far bottom of the figure "hatching time" is mentioned, but it was never measured in this study. Please, delete that piece of information.

Line 102: Delete "selected";

103: Please, describe the concentrations tested;

Line 104: Replace "each pesticide test" with "each tested pesticide";

Line 104: Replace "and diluted with distilled water" with "with distilled water used as solvent";

Lines 103-107: Did you use spray bottles or a Potter tower for spraying? Either way, give more details about the spraying method such as pressure, distance of application, volume applied, volume that actually hit the eggs, room conditions (temperature and humidity), etc.

Line 105: Replace "eggs" with "egg";

Lines 105-106: Replace "through direct contact by using spray droplets" with "by topical application using spray bottles" (assuming that you used spray bottle instead of a Potter tower);

Line 106: Replace "35ml" with "35 mL". What does that "35 mL" means? Also, what do you mean by "with graduated"?

Line 108: Was 0.15 mL the volume that actually hit the eggs? How did you calculate that?

Line 108: Delete "of the pesticide solution or distilled water for the control";

Line 110: Which "hatching condition"? Also, please describe the age of the eggs used in the experiment;

Lines 112-113: Replace "To compare the toxic difference between these pesticides, we calculated the index of relative toxicity which were" with "The pesticides were compared by the index of relative toxicity,";

Lines 113-114: The calculus that is described is not clear to me. Could you provide more details and also explain why you chose this method?

Lines 118-119: The description of the fourth grade ("extremely high risk", I suppose) is missing;

115-119: Make it a single paragraph;

Line 133: Delete "the"; replace "H. axyridis" with "Harmonia axyridis (Coleoptera: Coccinellidae)".

Table 1: Is it really necessary to have a column for the manufacturers? Maybe delete that column?

Line 137: Replace "temperature" with "fluctuating temperature";

Line 137: Weak? Was there any correlation at all? 

Line 138: I would say that FTR temperature had no (not less) impact on egg hatching;

Line 139: Replace "thermal" with "thermal fluctuations";

Line 142: Please indicate the actual P-value;

Line 143: Replace "were showed" with "are shown";

Lines 156-157: Please, give more details about the analysis, experiment, and the species you are studying; avoid abbreviations;

Line 158: I think this statement can either be deleted or transferred to somewhere in the line 125;

Lines 162-163: Please, give more details about the analysis, experiment, and the species you are studying; avoid abbreviations;

Fig. 2 and Table 3: The result for CLT is represented by a green dot in Fig. 2, which according to the figure's legend means that its hatching rate is in between 60 and 80%. However, in Table 3, the value for CLT is 54.29%, which in Fig. 2 is classified as color light blue. Please, find where the mistake is and correct it;

Table 3: This table must be redone to show the results for temperature x frequency only, because according to Table 2 this is the only significant interaction. Obviously, the statistics must be rerun accordingly. Please, pay attention to the letters used to differentiate the means, many of them are missing and/or misplaced in the current table;

Lines 174-175: Please, replace "resulted in the lowest hatching rate" with "showed the highest toxicity to eggs";

Lines 188: Please, give more details about the analysis, experiment, and the species you are studying; avoid abbreviations;

Lines 195-200 and 202-207 should be in the Introduction, not here in the Discussion section.

Table 4: Please, explain the reason for that line separating the 20% Acetamiprid SL from the other treatments. In the second column (regression equation), please put all the equations in the slope-intercept model (y = mx + b). The equation for 0.3% azadirachtin SC appears to have a mistake, please check it. 

Lines 204-207: The text is confusing. Please, make it clearer.

Line 205: What is "M. rotundata"? This is the first time that this species was cited within this manuscript, its scientific name should not be abbreviated. Also: include the order and family of that species following its non-abbreviated scientific name.

Lines 209-220: Which "egg cards" and "corps" are you referring to?

Line 209: Please, replace "in field" with "in the field";

Lines 219-226 should be in the Introduction, not here in the Discussion section;

Lines 233-234: Replace "20% acetamiprid SL" with "1.3% matrine EW". Also, include 20% Imidacloprid SL in your discussion since it was classified as high-risk as 20% acetamiprid;

Line 235: Include 30% Thiamethoxam SC in you discussion since it was as low-risk as 1.3% Matrine;

Lines 236-237: Where is this data? I did not see it in the Results section;

Lines 237-238: You mean the larva contacted insecticides residues following hatching? Explain better;

239-240: You mean the impact of residues left on the egg shells on the survival and biology of the larvae that hatch from those insect-contaminated eggs? Explain better.

Lines 247-249: Please, explain how;

Lines 249-250: Your study did not explore this question nor you mentioned it or included any citation on it earlier in the text. Hence, this statement seems lost here. I suggest deleting it.

Lines 261-262: What is this reference doing here?

Other comments: 

Please, stick to a single term when referring to a given variable. Using different terms confuses the reader. Three examples where this happened: (i) "fluctuating thermal regimen (FTR)" is interchanged with "recovery temperature" and also used to refer to the tested temperatures (15, 20, and 25 ºC), when in fact it refers to 36 different combinations of temperatures, times, and frequencies; (ii) "recovery duration" is interchanged with "exposure time", "thermal time", and "recovery time"; and (iii) "recovery frequency" is interchanged with "heating interval", "fluctuation frequency,"heating frequency" and is also called "thermophase recovery frequency", making it look like an specific type of recovery frequency, when in fact it is just the same old recovery frequency.

I miss the data on egg development time (or incubation time) and other secondary effects of FTRs and pesticides on the ladybird biology (=quality). These are important pieces of data that would allow us to better conclude whether the FTR and/or pesticide treatment can be actually safe for the biocontrol agent. 

Where are the figure labels?

The Methods section must be improved to provide greater detail about the experimental procedures and data collection.

The Discussion section is extremely weak and mainly explores two wrong paths. First, it is greatly based on justifying and highlighting the importance of the study by including citations from several other studies. Twenty out of the 58 lines of text (=34%) included in the Discussion are dedicated to this. Second, it repeats information already described in the Results section. This was done in 17 lines, i.e., 29% of the Discussion section. Additionally, there is data in the Discussion that was not presented in the Results section, as well as mistakes done by pure lack of attention to detail, such as when 20% acetamiprid was considered the least toxic insecticide when in reality it showed the highest index of relative toxicity. The Discussion section must be significantly improved to actually discuss the data, i.e., interpret the results and draw conclusions from them; as well as connect those results and conclusions to the literature to draw other conclusions that are not possible when each part is considered separately.

I recommend English revision by a native English speaker

Reviewer 2 Report

Dear authors, in the document you will find the suggestions.

The document requires that a native English speaker review it. In some parts it is difficult to understand the wording.

Reviewer 3 Report

This is an interesting study and is relevant to the successful use of this coccinellid predator for biocontrol.  However, the manuscript suffers from lack of clarity and consistency in the description of the methods, the terms used in the experimental design and a lack of clear correspondence between design, analysis and results.  Key questions and comments:

lines 89-92. description of FTR is not clear; what is a "recovery" and what exactly is meant by the FTR temperatures of 15, 20 and 25?  Fig. 1 is somewhat helpful but not thorough enough.  Text is simply not clear.

line 113 - what is a "standard pesticide"?

line 115 - what is a safety factor? Producing a ratio of an LC95 for what test insect with which field concentration?

line 127 - what is "inhibition probability of egg hatching"?

In Fig. 2, the labels for axes are not clear and do not clearly match the language used in describing the factors of the experimental design.  e.g. what is "exposure time"??

Interactions between factors are best shown with line graphs, not tables or scatterplots 

In summary, there is a lot of very unclear and inconsistent language and terminology that leads to a lot of difficulty in understanding the methods and the results.

Overall, the language is quite good.  However, sentence structure needs work in multiple parts of the manuscript.  For example, in line 62 the verb "are" is missing.  The main problem is the use of inconsistent, incomplete and unclear descriptions of methods, factors and the connection to the description of the results. 

Round 2

Reviewer 2 Report

Dear authors, review again the manuscript "Effects of fluctuating thermal regimes and pesticides on egg hatching of a natural enemy Harmonia axyridis (Coleoptera: Coccinellidae)"

I have noticed a substantial improvement in all sections of the manuscript.

Regarding the English language of the manuscript, it requires a minor edition.

Author Response

Thanks for your positive comments. We also appreciate your suggestion. We have carefully checked the linguistic issue throughout the manuscript. The revised manuscript has been uploaded to the system.

Reviewer 3 Report

Authors have made improvements to clarify terminology and methodology.  They still need to make the Figure legend complete for Fig. 2, as it is not acceptable to just label it as "Egg hatching rate under different level of three factor."  The three factor(s) must be described and interaction pointed out.  As I said before, and I urge the authors again, a line graph to show how lines cross each other is the best way to illustrate interactions between factors.

English is OK.

Author Response

Thanks for your positive comments. We also appreciate your suggestion. Based on your feedback, we have revised Figure 2 to better present our results. For more details see the newly submitted manuscript.